# Introducing Biomedisa as an open-source online platform for biomedical image segmentation

Philipp D. Lösel [1,2 ✉], Thomas van de Kamp [3,4], Alejandra Jayme[1,2], Alexey Ershov[3,4], Tomáš Faragó[3], Olaf Pichler[1,5], Nicholas Tan Jerome[6], Narendar Aadepu[7,8], Sabine Bremer[3,4,7], Suren A. Chilingaryan[6], Michael Heethoff[9], Andreas Kopmann [6], Janes Odar[3,4], Sebastian Schmelzle[9], Marcus Zuber [3,4], Joachim Wittbrodt [7], Tilo Baumbach[3,4] & Vincent Heuveline[1,2,5]

We present Biomedisa, a free and easy-to-use open-source online platform developed for semi-automatic segmentation of large volumetric images. The segmentation is based on a smart interpolation of sparsely pre-segmented slices taking into account the complete underlying image data. Biomedisa is particularly valuable when little a priori knowledge is available, e.g. for the dense annotation of the training data for a deep neural network. The platform is accessible through a web browser and requires no complex and tedious configuration of software and model parameters, thus addressing the needs of scientists without substantial computational expertise. We demonstrate that Biomedisa can drastically reduce both the time and human effort required to segment large images. It achieves a significant improvement over the conventional approach of densely pre-segmented slices with subsequent morphological interpolation as well as compared to segmentation tools that also consider the underlying image data. Biomedisa can be used for different 3D imaging modalities and various biomedical applications.

[1] Engineering Mathematics and Computing Lab (EMCL), Interdisciplinary Center for Scientific Computing (IWR), Heidelberg University, Im Neuenheimer Feld 205, 69120 Heidelberg, Germany. [2] Heidelberg Institute for Theoretical Studies (HITS), Schloss-Wolfsbrunnenweg 35, 69118 Heidelberg, Germany. [3] Institute for Photon Science and Synchrotron Radiation (IPS), Karlsruhe Institute of Technology (KIT), Hermann-von-Helmholtz-Platz 1, 76344 Eggenstein-Leopoldshafen, Germany. [4] Laboratory for Applications of Synchrotron Radiation (LAS), Karlsruhe Institute of Technology (KIT), Kaiserstr. 12, 76131 Karlsruhe, Germany. [5] Heidelberg University Computing Centre (URZ), Im Neuenheimer Feld 293, 69120 Heidelberg, Germany. [6] Institute for Data Processing and Electronics (IPE), Karlsruhe Institute of Technology (KIT), Hermann-von-Helmholtz-Platz 1, 76344 Eggenstein-Leopoldshafen, Germany. [7] Centre for Organismal Studies Heidelberg (COS), Heidelberg University, Im Neuenheimer Feld 230, 69120 Heidelberg, Germany. [8] Institute of Biological and Chemical Systems (IBCS), Karlsruhe Institute of Technology (KIT), Hermann-von-Helmholtz-Platz 1, 76344 Eggenstein-Leopoldshafen, Germany. [9] Ecological Networks, Technical University of Darmstadt, Schnittspahnstr. 3, 64287 Darmstadt, Germany. ✉email: philipp.david.loesel@gmail.com

Three-dimensional imaging is driving progress in many scientific disciplines. The analysis of volumetric medical and biological imaging data from e.g., X-ray computed tomography (CT), magnetic resonance imaging (MRI) or optical microscopy often requires isolating individual structures from the 3D volume by segmentation. Ongoing improvements in imaging technologies result in higher resolutions and faster acquisition times[1,2], hence increasing the demand for accelerated image analysis. Especially image segmentation is still a major bottleneck and often the most labor-intensive and error-prone task of 3D image analysis. One promising route towards faster segmentation builds on recent progress with deep neural networks[3]. However, the performance depends on large amounts of (usually) manually segmented training data and, like other automatic methods, its application is limited to repetitive structures such as certain organs[4], tumors[5], cells[6] or model organisms[7,8].

In situations where little a priori knowledge is available, fully automatic segmentation routines are less feasible. In these cases, manual segmentation by an expert followed by morphological interpolation remains a very common approach[9–12]. Here, labels are assigned to various structures of interest with different intervals inside the 3D volume (depending on the complexity of the dataset), followed by an interpolation of the labels between the pre-segmented slices. The underlying image data are usually not taken into account and the interpolation is therefore based exclusively on the segmented slices. Consequently, only a fraction of the real experimental information is utilized to derive the segmentation. Several commercial and free software packages for 3D data segmentation support such a conventional morphological interpolation[13,14], e.g. MITK[15], ITK-SNAP[16], ImageJ/Fiji[17], 3D Slicer[18], Microscopy Image Browser[19], Amira/Avizo, MeVisLab and Dragonfly.

Ensuring proper 3D segmentation of complex samples based on the morphological interpolation of pre-segmented 2D slices, dense pre-segmentation is required, sometimes even slice-by-slice. Therefore, the conventional approach to manual segmentation of 3D images is often tedious and time-consuming, effectively impeding the analysis of large amounts of data from samples of high morphological variability e.g. as required for the digitization of scientific collections or quantitative studies on biodiversity. Furthermore, artifacts resulting from morphological interpolation of manually segmented slices and subsequent correction (e.g. line artifacts, overly smooth meshes etc.) limit the quality of the results.

Semi-automatic image segmentation has been widely used in various applications. There are many types of initialization and user interaction, e.g. contour-[20,21] and bounding box-based[22] methods. Initializing the segmentation by seed points is particularly popular, e.g. in Graph Cuts (GC)[23], GrowCut[24], GeoS[25,26], Watershed[27] and Random Walker (RW)[28]. Machine learning applications such as the Trainable Weka Segmentation[29], ilastik[30] and Slic-Seg[31] can achieve good segmentation results, but are limited to application to distinctive structures such as cells, fibers, etc. and to user-defined features that depend on experience. Convolutional neural networks (CNNs) have also been used for semi-automatic segmentation[32–34]. However, like CNNs in general, they require fine-tuning of the hyperparameters for optimal performance, lack generalizability to previously unseen objects, are hard to configure, and their training is time consuming. Overall, the high level of complexity of neural networks hinders a fast solution for novel scenarios and diverse image data.

Instead of morphological interpolation, interactive methods can also be used for interpolation between pre-segmented slices, which are considered as seed points. However, the correct parameterization and application of these methods is a major challenge for life science researchers without in-depth IT knowledge.

Here, the implementations of GeoS in GeodisTK, GC in MedPy, and RW in scikit-image[35] are particularly popular, but using these tools requires programming skills. In addition, the implementations of GC and RW depend on parameters that have to be supplied manually based on experience or "trial and error", since their ideal settings differ greatly from application to application. Furthermore, both are limited by the image size due to their high memory requirement. This also applies to the GrowCut implementation in 3D Slicer. The active contour segmentation method integrated in the interactive software tool ITK-SNAP also requires the manual setting of parameters. Additionally, only one object can be segmented at a time, which increases the effort for the user when segmenting an image with multiple labels. These obstacles prevent many scientists in biology and medicine from using these techniques. Instead, they opt for traditional interpolation methods, i.e. manual labeling of slices, followed by morphological interpolation without considering the underlying image data.

With the goal of reducing the effort for the human annotator when segmenting large and complex samples of unknown composition, we developed Biomedisa (Biomedical Image Segmentation App, https://biomedisa.info/), an intuitive and freely available online platform that is easily accessible through a web browser and does not require any software installation or maintenance when used online. Biomedisa aims to be a one-button solution without a complex and tedious configuration that meets the needs of scientists with no substantial computational expertise.

Biomedisa uses weighted random walks[36] for smart interpolation, taking into account both the pre-segmented slices and the entire original volumetric image data. The method works well without parameter optimization and was specifically developed for massively parallel computer architectures such as graphics processing units (GPUs) in order to cope with constantly growing image sizes. Since the speed of single-core implementations is limited by the CPU frequency, the development of which has been slowed down due to physical limitations, image processing algorithms increasingly benefit from parallelization. Since they are independent of each other, the calculation of Biomedisa's random walks can be largely parallelized. They are therefore ideally suited to be calculated with GPUs and thus for processing large volumetric images.

In addition, Biomedisa offers several post-processing functions (see "Methods"). These include the ability to remove outliers or fill holes, to smooth the surface, to post-process the result with active contours, and to quantify the uncertainty with which the result was obtained. Furthermore, the data can be visualized with a slice viewer or 3D rendering software and shared with other users.

In medical imaging the application of deep learning techniques becomes increasingly promising to solve various segmentation or classification tasks. The key challenge is the availability or creation of a large amount of annotated ground truth data. Biomedisa particularly supports the dense annotation of training data for a deep CNN. Instead of an artificial augmentation to increase the training data, Biomedisa provides an easy way to extend the available sparsely annotated training data, thus providing a large amount of high-quality labels. Biomedisa can also be used to train a CNN for fully automatic segmentation when segmenting a large number of similar structures, e.g. human hearts or mouse molar teeth (see "Methods").

## Results

**Application**. We demonstrate the Biomedisa work flow and performance on a volumetric synchrotron X-ray microtomography (SR-μCT) dataset of a *Trigonopterus* weevil[37,38] with a size of 1497 × 734 × 1117 voxels (Figs. 1–4 and Supplementary

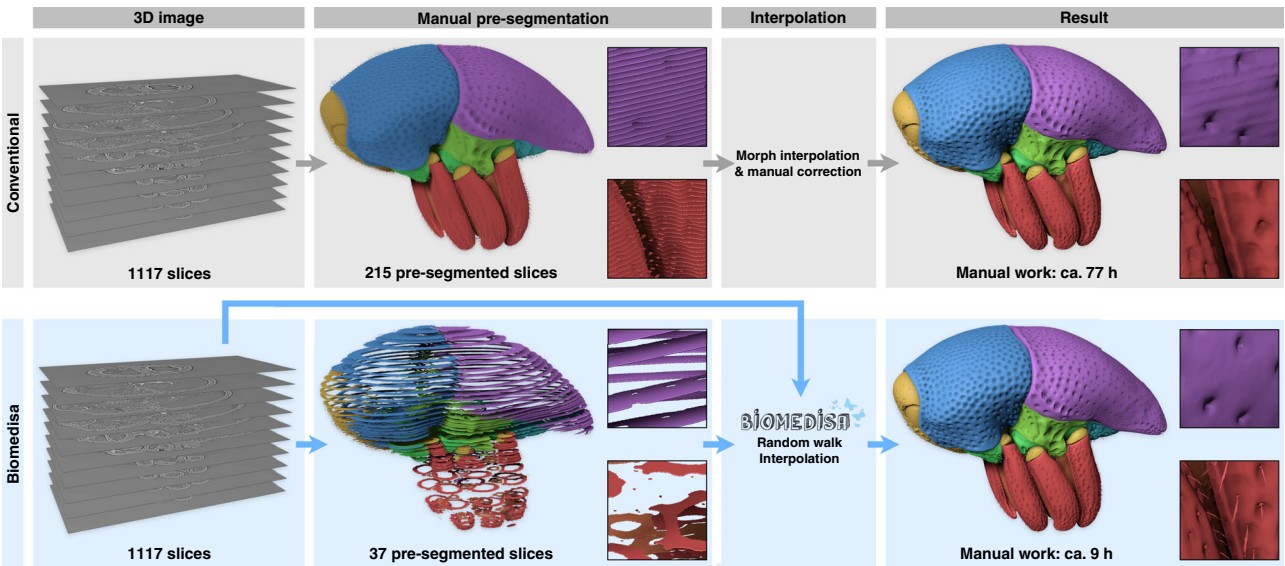

**Fig. 1 Comparison between a conventional segmentation approach and Biomedisa.** Both procedures require manual pre-segmentation of the 3D image stack. While the widely used morphological interpolation solely considers labels on pre-segmented slices, Biomedisa takes both the underlying 3D image data and the pre-segmented slices into account, resulting in a significantly lower amount of required manual input. Moreover, interpolation artifacts are avoided and fine details like hairs, which are usually omitted during manual segmentation, are included.

Movie 1). In order to create an interactive 3D reconstruction consisting of 64 individual body parts and for comparison with a conventional approach, the 3D image stack was processed both with a morphological interpolation, i.e. without utilizing underlying tomographic data, and with Biomedisa. Prior to morphological interpolation, every fifth slice (equaling 215 slices) was pre-segmented to ensure a decent result, which still required extensive manual correction. In total, it took about 77 h to obtain the final segmentation result (52 h for manual pre-segmentation and 25 h for correction of the interpolation result). In contrast, using Biomedisa, only 37 pre-segmented slices (specifically adapted to the weevil's morphology and equaling ca. 9 h of manual work) allowed for a precise final segmentation, even including fine surface details such as hair (Figs. 1, 2). The *Trigonopterus* dataset required less pre-segmented slices at the top (upper pronotum and elytra) and bottom (distal parts of the legs) than in the ventral thorax region with its tight articulations. The Biomedisa result was obtained by uploading both the image stack and the corresponding labels to a new project on Biomedisa and starting the semi-automatic segmentation with its default configuration. After the segmentation was completed, the result was downloaded and processed with Amira 5.6 and CINEMA 4D R20 in order to obtain the renderings of the figures and animations.

Biomedisa has been extensively tested and successfully applied to a wide range of 3D datasets from µCT, SR-µCT and MRI. Examples include a fossil wasp in amber[39], a theropod dinosaur claw in amber, an ethanol-preserved bull ant queen, an ethanol-preserved hissing cockroach, a medaka (Japanese rice fish) scanned in agarose, human hearts[40] and mouse molar teeth[41] (see "Methods", Fig. 5 and Supplementary Movie 2). Moreover, it has been used in a large comparative study based on segmentation of delicate, highly diverse structures in variously preserved states of fossil parasitic wasps inside mineralized fly pupae[42].

**Biomedisa online platform**. Biomedisa is an online platform, which is accessible through a web browser and requires no complex and tedious configuration of software and model parameters, thus addressing the needs of scientists without substantial

computational expertise. It supports several features for semi-automatic and automatic image segmentation and can be extended by additional user-defined functions that can be called via feature buttons. Biomedisa is implemented using Python and built on the Django project. Tasks are processed by several queues in a computing cluster. When a compute node is busy, tasks are automatically queued or assigned to an inactive compute server.

**Weighted random walks for image segmentation**. Biomedisa's random walks are inspired by the traditional RW algorithm[28]; however, they differ considerably in their purpose, implementation, and calculation of the weights (see "Methods"). In RW, a system of linear equations is solved to calculate the probability that a random walk starting at a voxel first hits a particular label. In contrast to RW, Biomedisa's random walks are performed in a Monte Carlo scenario and performed on multiple GPUs (see "Methods"). They start in the pre-segmented slices and diffuse into the volume, where the weights of the random walks (see "Methods") depend on the image data. Over time, the voxels of the volume are hit by the random walks. The segmentation is then performed by assigning each voxel to the label from which most hits originate. Voxels that have never been hit are automatically assigned to the background. Each step of a random walk can be considered as a throw to a target with six different sized fields, where the six fields are given by the weights and correspond to the six directions a random walk can potentially take in a three-dimensional image (six-connected voxels). The larger a field or weight, the better a voxel matches the start position of the random walk. For parallelization, each thread is assigned to a pixel in the pre-segmented slices and calculates all random walks that start from this pixel. The required random numbers are calculated using the Multiple Recursive Random Number Generator[43]. The computation is integrated into the calculation of the random walks for performance increases. The seeds of the first random walk are given by the index of the start position.

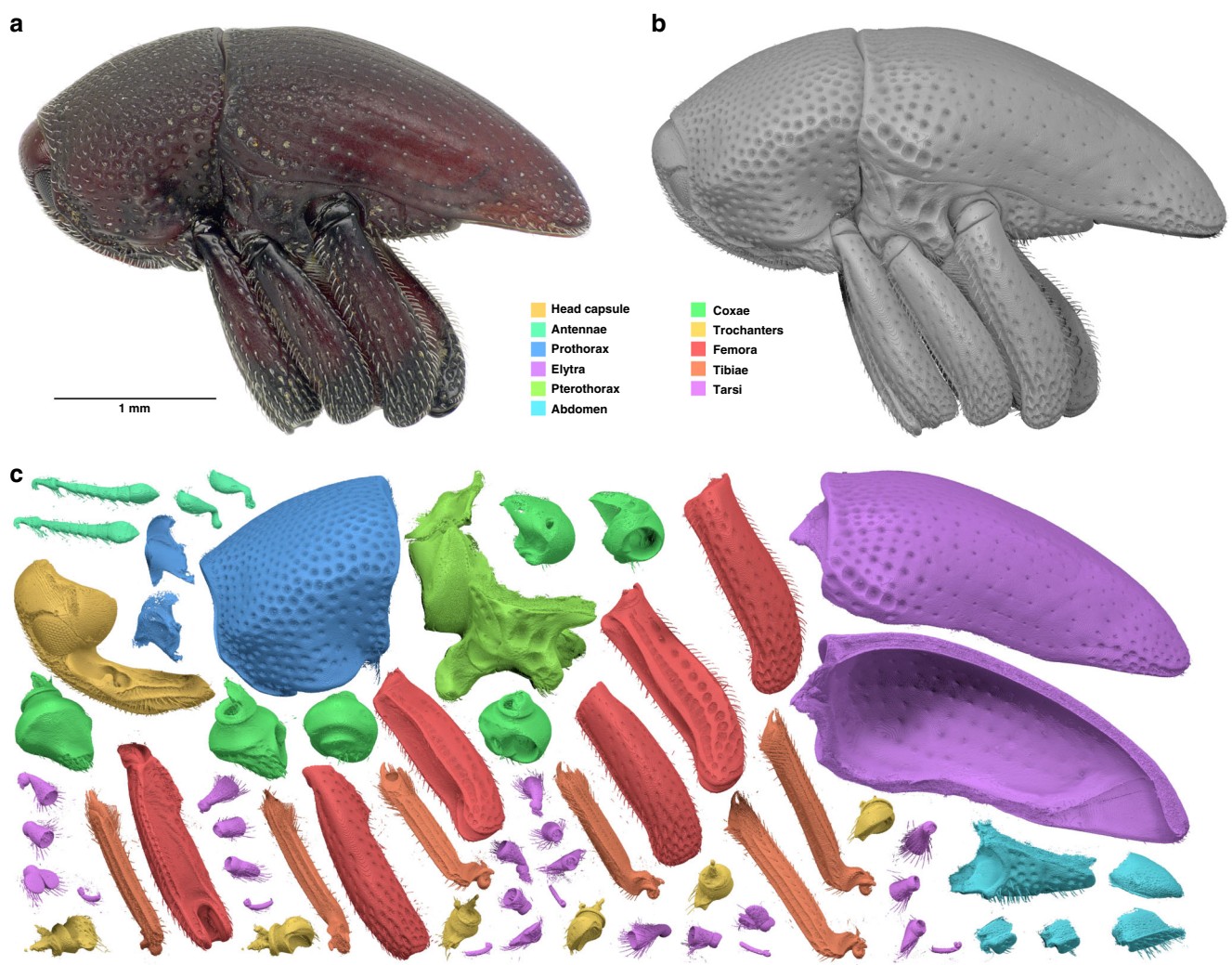

**Fig. 2 Biomedisa segmentation using a _Trigonopterus_ weevil as an example. a** Photograph of the original specimen. **b** Result of Biomedisa segmentation based on 37 pre-segmented slices of the tomographic volume adapted to the weevil's morphology. **c** The 64 isolated body parts of (**b**). The surface meshes shown in this figure are based on the original Biomedisa result. If necessary, outliers or minor flaws in the reconstruction (e.g. tiny holes) can be corrected with low effort.

**Adaptive random walks**. To significantly reduce computation time without impairing the quality of the result, the number of random walks is adapted based on the pre-segmented slices. For high-quality segmentation, the random walks near the edges of the object are of crucial importance, since these are in direct competition with the random walks of the adjacent label. Random walks that start far within the object barely compete with other random walks. Therefore, a reduced number of random walks are used in the inner area to accelerate the calculation. A start position is considered to be inside if there is no other label in an area of $101 \times 101$ pixels surrounding the start position. With equivalent results, the computation of the datasets bull ant queen, cockroach, theropod claw, _Trigonopterus_, mineralized wasp, and wasp from amber was 45% faster on average with adaptation.

**Smoothing**. To smooth the segmentation result while preserving fine structures such as hairs of insects, a specifically developed smoothing technique has been integrated (see "Methods"). While common morphological smoothing techniques use dilation and erosion, this method also considers the number of hits of the random walks, and thus implicitly the image data. This counteracts the disappearance of tiny but essential structures when

these areas are hit by many random walks. The smoothed result is offered as an optional segmentation result.

**Uncertainty of the result**. By quantifying the uncertainty (see "Methods"), users receive feedback on the quality of the segmentation result. The uncertainty result highlights areas that should be corrected manually or which pre-segmented slices should be revised before restarting the process. The uncertainty considers the influence of poorly pre-segmented input data and problematic image areas (e.g. filigree structures, lack of contrast or image artifacts) on the result. The segmentation is considered uncertain when random walks compete with random walks from other labels without superior candidates, i.e. when the number of hits is approximately the same. Conversely, the probability for a correct assignment of the voxel is considered high if the hits of random walks coming from a label clearly dominate or if the voxel is only hit by random walks from a single label. The Biomedisa results of the _Trigonopterus_ dataset show the same level of uncertainty for an initialization with 215, 108, 54 (corresponding to a pre-segmentation of every 5th, 10th and 20th slice) and 37 pre-segmented slices (adapted to the weevil's morphology) (Fig. 4). The uncertainty of the results for 27 (pre-segmentation of

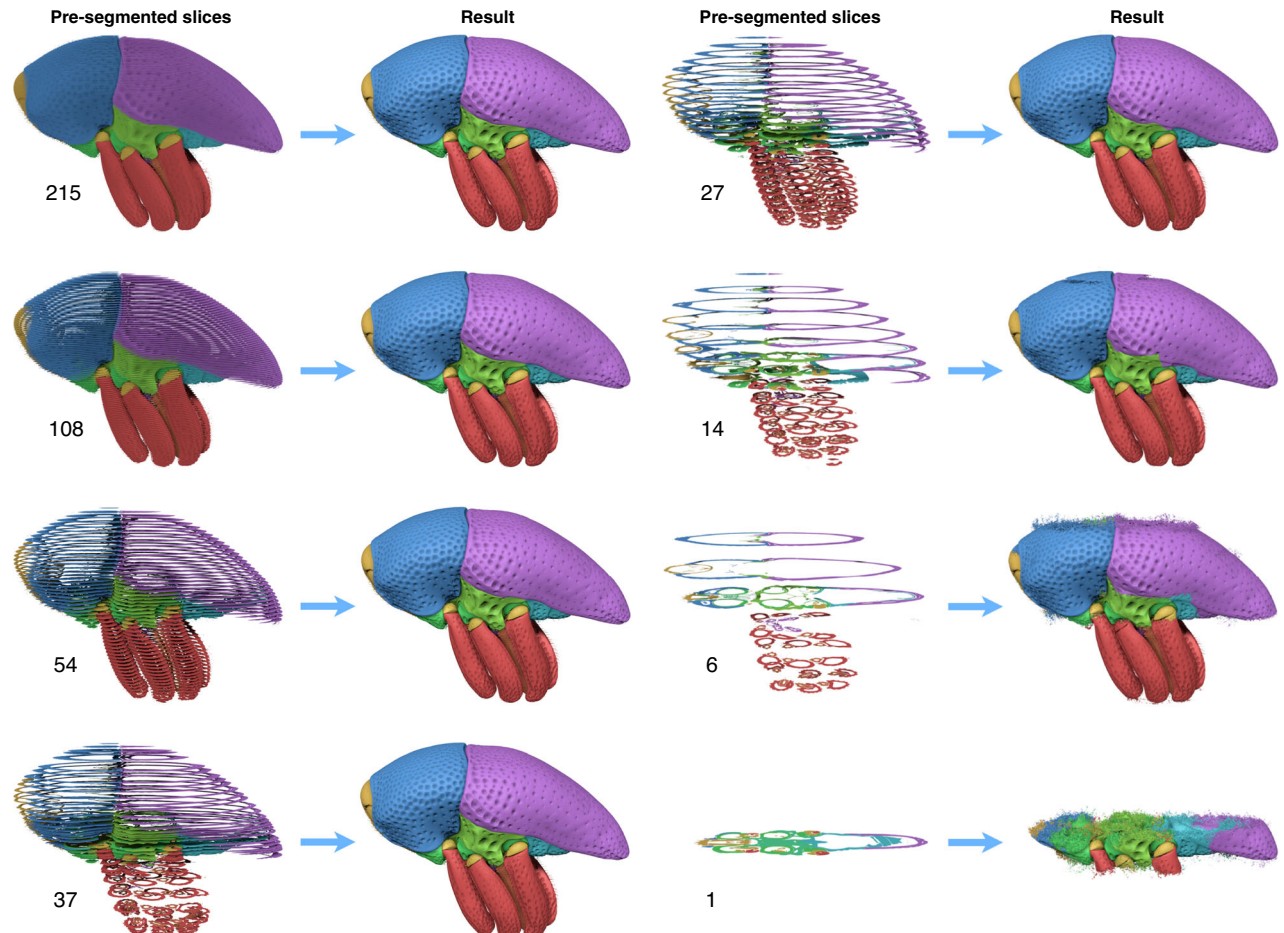

**Fig. 3 Biomedisa results based on different numbers of pre-segmented slices.** Inputs of 215 (as used for morphological interpolation), 108 and 54 equally spaced slices that correspond to pre-segmentation of every 5th, 10th and 20th slice provided accurate results. By adapting the spacing between the slices to the weevil's morphology, a much lower count of only 37 slices yielded a dataset of equal quality. Lower numbers of pre-segmented slices resulted in increasingly flawed outputs.

every 40th slice) or fewer slices increases significantly. In all cases, the boundary between elytra and thorax, which are closely interlocked, is highlighted in the volume renderings of the uncertainty. The boundary is almost invisible in the tomographic scan (Supplementary Fig. 1), which results in a high degree of uncertainty in the segmentation result.

**Convolutional neural networks for image segmentation.** Biomedisa supports training of CNNs for segmentation of three-dimensional image data. We integrated a very easy-to-use 3D U-Net[44], whose standard configuration achieves good results on our test datasets (see "Methods"), eliminating the need for a tedious optimization of hyperparameters. A network can be trained on segmented image data by selecting a set of image files along with the corresponding label files. After completing the training, the segmentation is carried out by selecting the images to be segmented together with the trained network. In addition, a second network can be trained to refine the segmentation results of the first network (see "Methods").

**Visualization.** A web-based visualization platform[45] is integrated into Biomedisa to enable a quick preview of the datasets. The visualization framework supports shader-based volume ray-casting that enables iso-surface and volume renderings. These rendering modes are essential to identify the inner or outer structure of the sample. The input data use a slice map where tomographic slices are arranged in a gridded mosaic format. The framework emphasizes interactive visualization by varying the ray-casting sampling step where a higher ray-casting step provides better visual resolution at the expense of performance. In Biomedisa the ray-casting sampling step is kept low during user interactions such as rotation or zooming and set to a higher value in idle mode. The integrated 2D slice viewer allows for a quick impression of the result. For this purpose, a color is selected for each label. The edges of the labels are then highlighted as a contour in the image data.

**Data file formats and data types.** The following three-dimensional data file formats are supported: Multipage TIFF, Amira mesh (AM), MHD, MHA, NRRD and NIfTI (.nii and .nii.gz). In addition, a zipped folder containing two-dimensional slices as DICOM, PNG, or TIFF that represents the volume can also be uploaded. The result retains the meta information of the label file, e.g. label names and colors, so it can be easily re-imported and post-processed in the user's preferred segmentation tool. Data can be processed in integer or float as 8-bit, 16-bit, 32-bit or 64-bit. There are two data processing procedures: To reduce memory usage, 8-bit images are processed separately, while 16-bit, 32-bit and 64-bit images are converted to single-precision floating-point format and scaled to the interval from 0 to 255. Before starting the segmentation, images can be converted to an 8-bit Multipage TIFF. This is useful if the available GPU memory

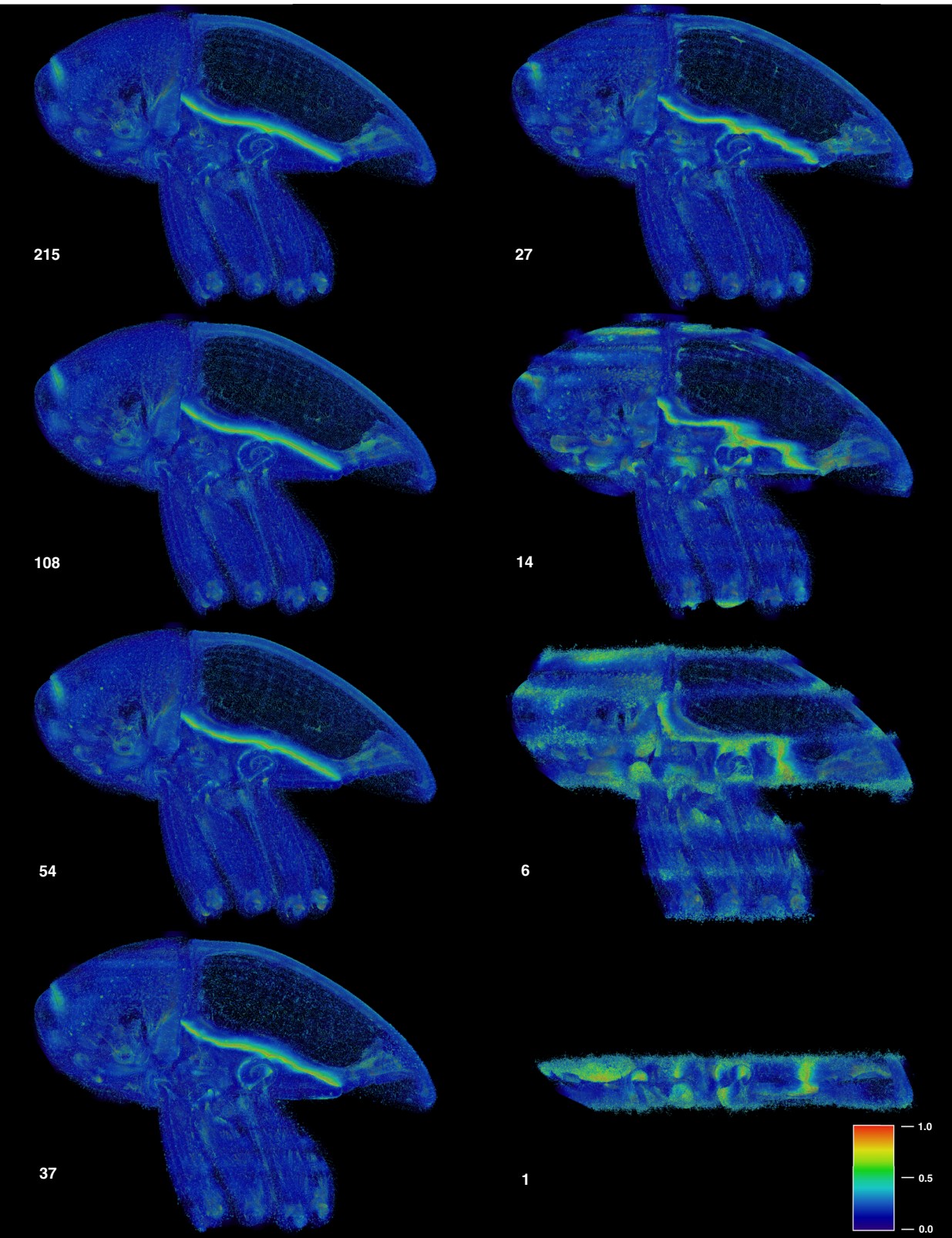

**Fig. 4 Uncertainty of Biomedisa results based on different numbers of pre-segmented slices.** For 215, 108, 54 and 37 pre-segmented slices, the results show approximately the same degree of uncertainty, while for 27 or fewer slices the uncertainty increases significantly. The conspicuous bright line represents the boundary between elytra and thorax, which are closely interlocked. The boundary is almost invisible in the tomographic scan, thus resulting in a high uncertainty of the segmentation result. The uncertainty values range from 0 (blue) to 1 (red), with 0 meaning no uncertainty and 1 meaning a high degree of uncertainty, i.e. a voxel can be assigned to at least two labels with the same probability.

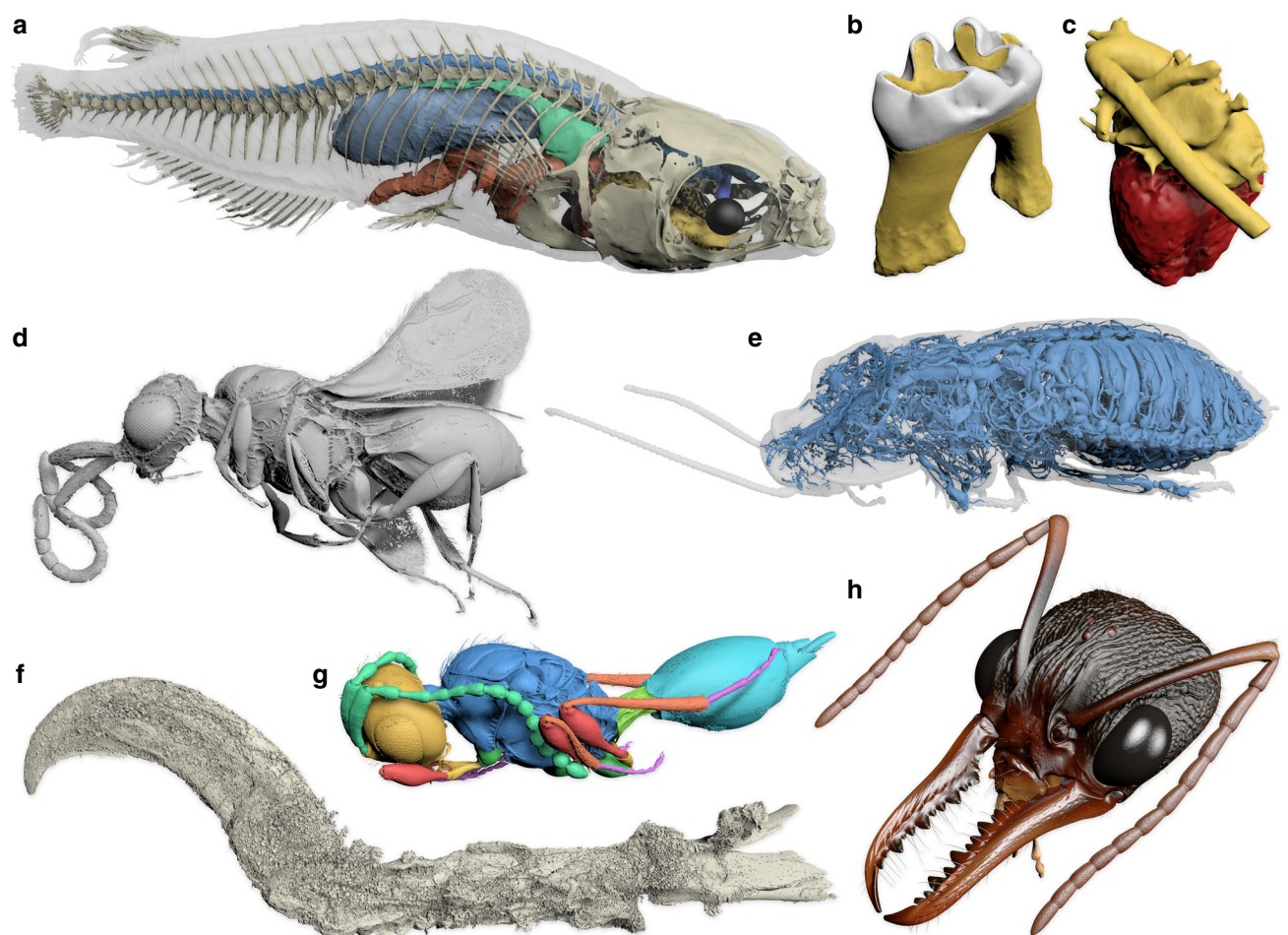

**Fig. 5 Biomedisa examples. a** Medaka fish with segmented skeleton and selected internal organs (based on μCT scan). **b** Mouse molar tooth showing enamel (white) and dentine (yellow) (μCT). **c** Human heart with segmented heart muscle and blood vessels (MRI). **d** Fossil parasitoid wasp from Baltic amber (SR-μCT). **e** Tracheal system of a hissing cockroach (μCT). **f** Claw of a theropod dinosaur from Burmese amber (SR-μCT). **g** Fossil parasitoid wasp preserved inside a mineralized fly pupa (SR-μCT). **h** Head of an Australian bull ant queen (SR-μCT). See "Methods" and Supplementary Table 2 for details on the specimens.

is insufficient to handle very large datasets, for example large 32-bit images. Data can be shared with other users by entering their usernames. Additionally, a password-protected download link can be created.

**Evaluation**. To compare the segmentation performance of Biomedisa with popular semi-automatic segmentation software, we use the implementation of RW in scikit-image, the GC implementation in MedPy, the geodesic distance algorithm (GeoS) in GeodisTK and the purely morphological interpolations of ITK and Amira to segment a variety of datasets (Table 1 and Fig. 6). We use a system with 2 Intel Xeon Gold 5120 CPU (14 cores each) with a base clock of 2.2 GHz and a boost clock of 3.2 GHz, 750 GB RAM and 4 NVIDIA Tesla V100. The results of the Amira segmentation were carried out by an expert using a standard desktop computer. Although Biomedisa automatically uses all available GPUs, we only use a single GPU for comparison with the other CPU-based segmentation tools. Here, GC and GeoS are single threaded processes, while RW is linked with a multithreaded BLAS library and automatically uses additional cores if possible.

To solve the system of linear equations in RW, we use the Conjugate Gradient method and algebraic multigrid as preconditioner. GeoS is computed with the fast raster scan algorithm. The

use of both RW and GC is strongly limited by the image size. While they are usable for small volumetric images, e.g. the datasets human hearts, human mandibles and mouse molar teeth, the segmentation of *Trigonopterus* (1497 × 734 × 1117 voxels) or even larger datasets (e.g. mineralized wasp) is not possible with these tools. In order to reduce the considered image size and to enable a comparison with these techniques for large datasets, we therefore calculate the segmentation block-by-block between two pre-segmented slices and add additional blocks at the bottom and the top for extrapolation beyond the first and last pre-segmented slice, respectively. For GeoS, we use the same block-by-block strategy in order to reduce the computing time. Using a block-by-block strategy, it is not necessary to calculate the geodesic distance for all labels in the entire volume because not all labels occur in all blocks. While the interpolation of ITK and Amira can be carried out for the entire dataset and does not require a block-by-block strategy, extrapolation beyond the first and last pre-segmented slice is not possible here. Except for Amira, image volumes were automatically cropped to the region of interest before processing, with at least 100 voxels between image boundaries and pre-segmentation. The medaka skeleton and the wasp from amber datasets are too large to be processed with a single GPU and had to be split into two parts (upper and lower part), which were then processed individually and finally combined into a single volume.

**Table 1 Quantitative comparison of different semi-automatic segmentation tools for the segmentation of different datasets.**

| Dataset | Method | Dice (%) | ASD (pixels) | Time (min) |
|---|---|---|---|---|
| Mineralized wasp (56 labels, every 20th slice pre-segmented, 1077 × 992 × 2553 voxels) | Biomedisa | **95.98** | **0.458** | 30 |
| | GeodisTK (GeoS) | 94.21 | 0.583 | 332 |
| | Scikit-image (RW) | 79.42 | 4.196 | 372 |
| | MedPy (GC) | 88.29 | 3.803 | 1943 |
| | ITK interpolation | 79.96 | 3.008 | 10 |
| | Amira interpolation | 79.04 | 3.762 | 21 |
| *Trigonopterus* (64 labels, smart pre-segmentation, no cross-validation, 1497 × 734 × 1117 voxels) | Biomedisa | **97.81** | **0.382** | 20 |
| | GeodisTK (GeoS) | 96.99 | 0.553 | 415 |
| | Scikit-image (RW) | 80.79 | 6.001 | 494 |
| | MedPy (GC) | 36.13 | loss of 38 labels | 2419 |
| | ITK interpolation | 82.75 | loss of 4 labels | 11 |
| | Amira interpolation | 84.66 | loss of 3 labels | 13 |
| Wasp from amber (15 labels, every 40th slice pre-segmented, 1417 × 2063 × 2733 voxels) | Biomedisa | **95.95** | **0.762** | 48 |
| | GeodisTK (GeoS) | 93.55 | 1.751 | 1057 |
| | Scikit-image (RW) | 88.80 | 6.987 | 1815 |
| | MedPy (GC) | 90.80 | 8.136 | 5591 |
| | ITK interpolation | 80.78 | 5.234 | 28 |
| | Amira interpolation | 81.52 | 6,892 | 19 |
| Theropod claw (1 label, every 80th slice pre-segmented, 1986 × 1986 × 3602 voxels) | Biomedisa | **88.67** | **0.409** | 21 |
| | GeodisTK (GeoS) | 60.02 | 3.845 | 182 |
| | Scikit-image (RW) | 16.69 | 18.174 | 541 |
| | MedPy (GC) | 0.0 | 29.542 | 563 |
| | ITK interpolation | 69.97 | 4.318 | 36 |
| | Amira interpolation | 66.35 | 5.815 | 3 |
| Medaka skeleton (1 label, every 10th−80th slice pre-segmented, 900 × 1303 × 4327 voxels) | Biomedisa | **84.24** | **1.210** | 28 |
| | GeodisTK (GeoS) | 76.40 | 1.694 | 196 |
| | Scikit-image (RW) | 5.01 | 17.122 | 537 |
| | MedPy (GC) | 7.59 | 20.072 | 629 |
| | ITK interpolation | 39.04 | 7.220 | 17 |
| | Amira interpolation | 23.69 | 11.976 | 2 |
| Human hearts (2 labels, every 20th slice pre-segmented, 157 × 216 × 167 voxels on average) | Biomedisa | **90.96 ± 1.65** | 0.715 ± 0.210 | 3 ± 2 s |
| | GeodisTK (GeoS) | 89.58 ± 1.71 | **0.655 ± 0.176** | 13 ± 6 s |
| | Scikit-image (RW) | 81.37 ± 2.57 | 1.764 ± 0.307 | 52 ± 24 s |
| | MedPy (GC) | 88.36 ± 1.49 | 1.021 ± 0.223 | 61 ± 28 s |
| | ITK interpolation | 70.14 ± 3.60 | 3.468 ± 0.610 | 1 ± 1 s |
| | Amira interpolation | 68.67 ± 4.35 | 3.862 ± 0.618 | <30 s |
| Mouse molar teeth (3 labels, every 40th slice pre-segmented, 438 × 543 × 418 voxels on average) | Biomedisa | **98.39 ± 0.28** | **0.512 ± 0.072** | 1.3 ± 0.1 |
| | GeodisTK (GeoS) | 98.20 ± 0.20 | 0.585 ± 0.055 | 8.1 ± 0.6 |
| | Scikit-image (RW) | 81.89 ± 1.11 | 6.813 ± 0.493 | 17.4 ± 1.6 |
| | MedPy (GC) | 89.90 ± 2.44 | 6.620 ± 1.397 | 52.7 ± 6.4 |
| | ITK interpolation | 80.69 ± 1.93 | 6.375 ± 0.883 | 0.2 ± 0.1 |
| | Amira interpolation | 79.20 ± 2.25 | 6.651 ± 0.885 | <1 |

For the configuration, the values of the default parameters were chosen, i.e. $\beta = 130$ (RW), norw $= 10$ and sorw $= 4000$ (Biomedisa). Graph Cut and GeoS have no default values for $\sigma$ and the number of iterations, respectively. The values were therefore chosen from the examples in the documentation, i.e. $\sigma = 15$ (GC) and iterations $= 4$ (GeoS). If not explicitly stated otherwise, Dice and ASD scores are twofold cross-validation accuracies. If the dataset consists of several images, the standard deviation is given (±). Highest accuracy and best result are shown in bold font.

The segmentation results of RW and GC strongly depend on a parameter ($\beta$ and $\sigma$, respectively) which has to be supplied manually. The fast raster scan algorithm in GeoS only depends on the number of iterations. The more iterations, the better the result, but the longer the computation takes. Biomedisa's random walks depend on the number of random walks that start in a pre-segmented pixel (norw) and the number of steps of each random walk (sorw). Similar to GeoS, the more random walks are performed, the better the result, but the longer the calculation will take. Large distances between pre-segmented slices require a higher number of steps for each random walk. The default configuration of Biomedisa (norw $= 10$ and sorw $= 4000$) was selected empirically to cover a wide range of applications. For comparison, we use the standard configuration of all techniques ($\beta = 130$ (RW), norw $= 10$, sorw $= 4000$ (Biomedisa)) and if no default values are given, we use the configuration specified in the examples in the documentation ($\sigma = 15$ (GC), iterations $= 4$ (GeoS)).

We compare the tools on a variety of datasets based on different imaging technologies (MRI (human hearts[46]), μCT (mouse molar teeth, medaka fish), CT (human mandibles), and SR-μCT (*Trigonopterus*, mineralized wasp, wasp from amber, theropod claw)) (Table 1, Figs. 6, 7, Supplementary Figs. 1−13 and Supplementary Table 1), with a different number of segmented labels (ranging from 1 label (theropod claw and medaka skeleton) to 64 labels (*Trigonopterus*)), different image sizes (ranging from 157 × 216 × 167 voxels (average of human hearts) to 1986 × 1986 × 3602 voxels (theropod claw)) and different distances between the pre-segmented slices (ranging from every 10th to every 80th slice).

*Trigonopterus* was manually labeled every fifth slice by an expert (215 slices in total). The segmentation results for a pre-segmentation adapted to the morphology (37 slices, Table 1) and pre-segmentations of every 20th, 40th and 80th slice (corresponding to 54, 27 and 14 slices, Supplementary Table 1) were evaluated on the basis of the remaining pre-segmented slices, which are considered as ground truth.

For the datasets medaka (here only skeleton), theropod claw, mineralized wasp, wasp from amber, and mouse molar teeth, we

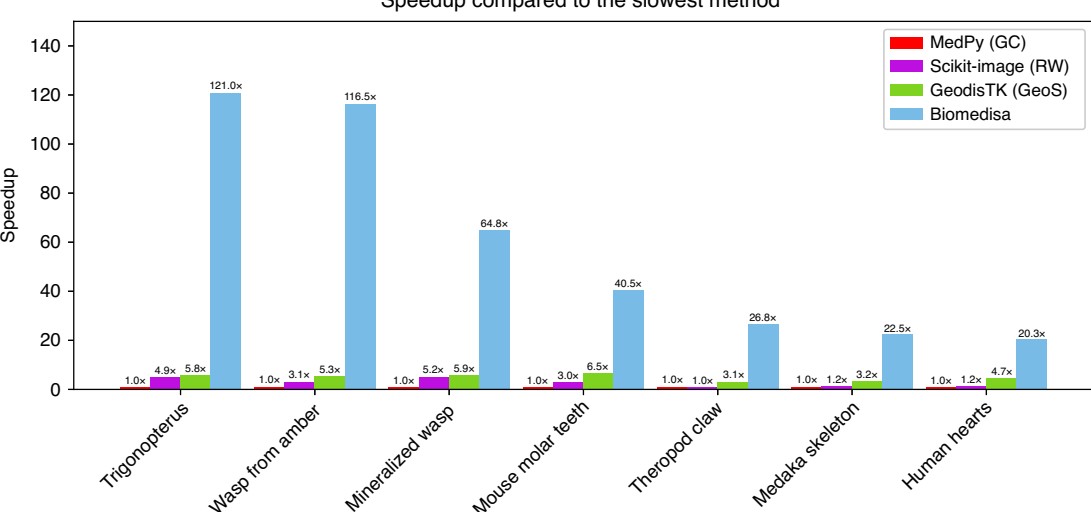

**Fig. 6 Speedup of computing times of GPU-based Biomedisa compared to CPU-based segmentation tools.** Speedup of computing times of different semi-automatic segmentation tools that take the image data into account compared to the slowest method according to Table 1. The values for mouse molar teeth and human hearts are average values.

use a twofold cross-validation, where we split the pre-segmented slices into two sets A and B, where A contains every second slice of the pre-segmented slices and B the remaining slices. First, we use the slices of A to initialize the algorithms and the slices of B to test the accuracy of the results. Then we use B for the initialization and A for the evaluation. Finally, the average of both accuracy scores is calculated to obtain the twofold cross-validation accuracy (Table 1). Consequently, the distances between the pre-segmented slices is double the size as used for creating the results shown in Fig. 5. For example, every tenth slice in the mineralized wasp was originally pre-segmented. For the cross-validation, we only use every 20th slice as initialization and the remaining pre-segmented slices for the evaluation (and vice versa). The human hearts have been fully manually labeled by a trained observer and validated by two clinical experts. Here, we extract every tenth slice of the manually labeled data to perform a twofold cross-validation.

To evaluate the accuracy of the segmentation results, we consider two metrics (see "Methods"), the Dice similarity coefficient (Dice) and the average surface distance (ASD). The Dice score quantifies the match of two segmentations and is between 0 and 1, where 0 means no overlap and 1 means a perfect match of two segmentations. The ASD is the average Euclidean distance from a point on the boundary of the pre-segmented slices (here considered as ground truth) to the closest point on the surface of the segmentation result. The smaller the ASD, the closer the segmentation result is to the ground truth. If at least one label is lost during the segmentation process, the ASD cannot be calculated. Instead, the number of lost labels is given here.

The results were evaluated without post-processing. Biomedisa achieves the best Dice score for all datasets and has the smallest ASD with the exception of human hearts (Table 1). The computation time for the segmentation varies considerably between the methods (Table 1). In all cases, Biomedisa is significantly faster than the methods that also take the image data into account (RW, GC, and GeoS). Even the computation time of the purely morphological interpolation of ITK and Amira is in the order of Biomedisa. Depending on the image size, the number of pre-segmented slices and the number of labels, the speedup of Biomedisa compared to GC is at least by a factor of 20 (human hearts) and up to a factor of 121 (*Trigonopterus*) (Fig. 6). In comparison to GeoS, the speedup ranges from 4 (human hearts)

to 22 (wasp from amber) and in comparison to RW, it ranges from 17 (human hearts) to 38 (wasp from amber).

In order to test the robustness of the methods against input errors, we use ten fully manually segmented human mandibles[47]. Each dataset was labeled by two human annotators resulting in two varying segmentations for each dataset (Fig. 7). About every 20th slice of these manual segmentations serve as initialization of the algorithms. Some manual segmentations have slices with partially or completely missing labels. An equidistant selection was therefore not always possible. In contrast to GeoS, RW, and GC, Biomedisa shows higher robustness against inaccurately pre-segmented slices (Fig. 7). Overall, Biomedisa's segmentation results based on the two varying initializations are more similar to each other (average Dice score of 98.83%) than the pre-segmented slices of the two human annotators (average Dice score of 94.62%) used to initialize the segmentation.

## Discussion

Biomedisa can significantly accelerate the most common segmentation practice for large and complex image data, i.e. the manual segmentation of densely pre-segmented slices and subsequent morphological interpolation, while at the same time improving the quality of the result (Fig. 1). Utilizing the complete 3D image information of the original data allows for much larger distances between neighboring pre-segmented slices compared to conventional manual segmentation, resulting in a considerably reduced amount of required manual work. The segmentation of the *Trigonopterus* weevil with Biomedisa required only 37 pre-segmented slices that were adapted to the morphology of the weevil and no correction of the result. Both drastically reduced the total amount of manual work compared to the conventional approach. Moreover, Biomedisa avoids interpolation artifacts, and fine details, which usually cannot be segmented properly by morphological interpolation, can be depicted correctly.

In addition, Biomedisa offers significant advantages over the compared CPU-based semi-automatic segmentation tools. Biomedisa is the only of all evaluated techniques that was specifically developed for parallel computer architectures. Due to their high scalability, Biomedisa's random walks perform well on GPUs (see "Methods"). Therefore, Biomedisa is considerably faster than the compared tools, which also consider the image data, especially for

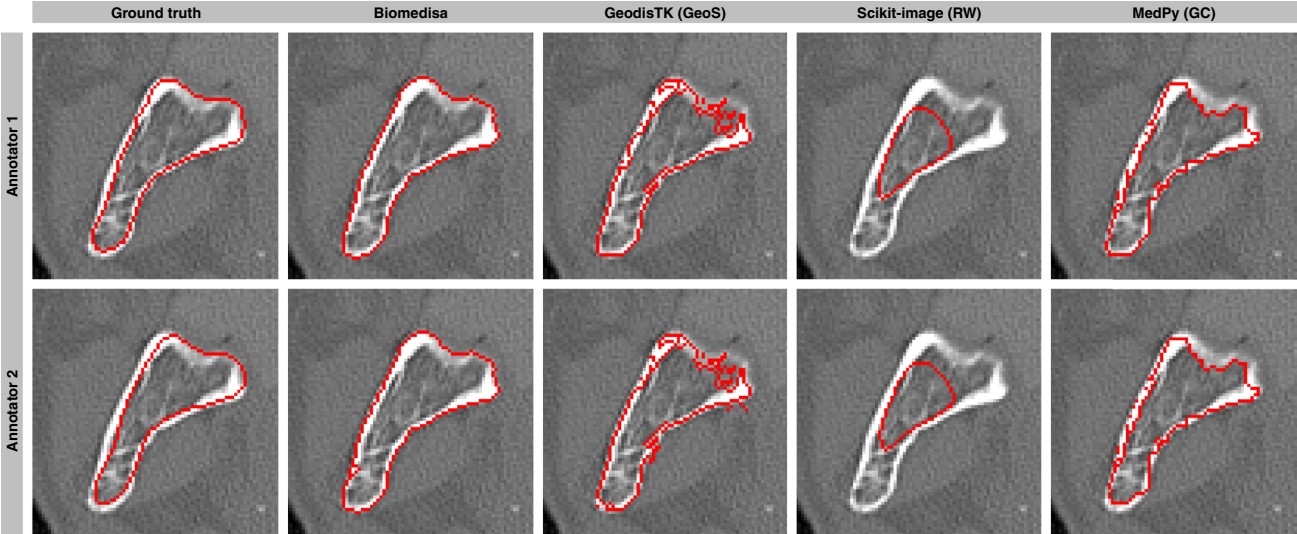

**Fig. 7 Visual comparison of robustness to input errors.** Visual comparison of different segmentation methods for segmenting a human mandible based on flawed and inconsistent ground truth data labeled by two annotators. For the configuration, the values of the default parameters were chosen. If no default values were given, the values were selected from the examples in the documentation. Here, every 20th slice of the manual segmentation was used as initialization. The images show the segmentation result between two pre-segmented slices.

large volumetric images. However, fast calculation is important because the pre-segmentation may need to be refined or additional slices added. Although a GPU-based implementation of GC, RW and GeoS might be possible, the development and implementation complexity of efficient parallel solutions is significantly higher than that of the existing CPU-based solutions. In addition, in contrast to these tools, Biomedisa's computing time does not change significantly with increasing number of labels, as the total number of random walks remains the same (Supplementary Table 1). Moreover, GPUs have steadily increased performance in recent years, which has led to a fundamental acceleration of Biomedisa, even without changing the code (Supplementary Fig. 14a). Although Biomedisa was developed for NVIDIA GPUs using CUDA and PyCUDA[48], the method can be implemented on any accelerator or multiprocessor.

Higher segmentation accuracies can be achieved without the need for complex and tedious configuration of unknown parameters. Biomedisa aims to be a one-button solution. Therefore, the evaluation of all techniques is based on their standard configuration. Two metrics were used for the evaluation (Dice and ASD). Both are among the most commonly applied metrics for evaluating performance in biomedical image segmentation challenges[49]. The evaluation was performed based on sparsely pre-segmented slices, which were provided by experts. Using a twofold cross-validation, only half of the pre-segmented slices were used to initialize the algorithms compared to creating the renderings in Fig. 5, which partly results in a low accuracy, especially for large distances between the pre-segmented slices, e.g. every 80th. In particular, the results of GC and RW depend highly on a parameter that has to be supplied manually. Therefore, the standard configuration of GC and RW work very poorly in most test cases. The results of both techniques might be optimized by properly configuring these parameters, but due to the high computing time, the necessary parameter optimization makes these techniques less feasible in practice for very large datasets. Compared to Biomedisa, GeoS struggles primarily with large distances between the pre-segmented slices and tiny structures such as hair (Supplementary Fig. 1). In addition to disregarding the underlying image data, another disadvantage of the purely morphological

interpolation of ITK and Amira is that extrapolation beyond the pre-segmented slices is not possible.

Additionally, Biomedisa is more robust against input errors. Biomedisa's random walks are able to slightly adjust inaccurately pre-segmented slices (Fig. 7), while the compared methods do not update pre-segmented slices. In the datasets of human mandibles, inaccurate but fixed pre-segmented slices disturb the convergence to a proper solution, especially for GC and RW.

Moreover, Biomedisa's online platform is extremely easy to use even without substantial computational expertise. In addition to the absence of parameter tuning, there is no need to install software or meet hardware requirements. Alternatively, the open-source version of Biomedisa can be installed on a desktop computer or in the local network of an institute.

Finally, with the support of many common data formats, Biomedisa can be easily used in tandem with common segmentation software, allowing the user to maintain their familiar 3D data analysis workflow and drastically speed up the process.

During its development, the random walk segmentation of Biomedisa was tested and already successfully employed in a number of studies[39–41,50,51]. It also played a crucial role in the description of parasitic wasps discovered in mineralized fly pupae[42]. The detailed segmentation of the wasps facilitated the species descriptions that would have been virtually impossible with a conventional manual segmentation approach.

It is generally recommended to start a Biomedisa project with a low number of pre-segmented slices adapted to the morphology. In contrast to GC, RW, and GeoS, which become slower the smaller the number of pre-segmented slices, Biomedisa becomes even faster the fewer slices are pre-segmented (Supplementary Table 1). This supports an interactive segmentation that begins with a small number of pre-segmented slices. If the result is flawed, additional slices can be added in the respective regions before restarting the process. This strategy is more efficient than the preemptive segmentation of many slices when it may not be necessary at all. For *Trigonopterus*, 37 pre-segmented slices (Dice score of 97.81%) yield results of comparable quality to a much higher number of pre-segmented slices, i.e. 108 (98.14%) and 54 (98.00%). Thereby the manual workload increases considerably with an increasing number of pre-segmented slices. For example,

using 108 slices instead of 54 doubles the manual workload from 13 to 26 h, while the Dice score increases only slightly by 0.14%. The result of 27 pre-segmented slices (97.28%) shows first minor flaws visible to the naked eye. A lower number of pre-segmented slices produces significantly flawed results (93.10% for every 80th slice, 81.37% for every 160th slice), but large parts of the weevil's morphology are still represented correctly (Fig. 3).

Biomedisa offers various features that enable semi-automatic and automatic image segmentation, reduced manual post-processing, and visualization of the segmentation result. The uncertainty feature shows the influence of missing or poorly pre-segmented slices and thus helps to correct and add slices in critical areas. Accessing neural networks on Biomedisa's online platform makes it easier for researchers with no deep learning experience to use CNNs, while advanced users can modify the source code and optimize hyperparameters. Additionally, Biomedisa can easily be expanded with additional features.

Our explicit aim was to create a freely available, user-friendly and widely applicable tool to improve the tedious manual segmentation procedure that still dominates 3D image analysis in many biomedical disciplines. Biomedisa was developed for CT and MRI but is generally suitable for many more types of volumetric image data, e.g. from confocal laser scanning microscopy, focused ion beam scanning electron microscopy or histological imaging. Even though the development of Biomedisa was motivated by applications from biology and medicine, it can also be employed for 3D data from other disciplines, e.g. geology, materials science and non-destructive testing.

## Methods

**Weights of Biomedisa random walks**. For each start position $x_0$ in a pre-segmented slice, the weights of the random walks that start in $x_0$ are computed by

$$w_{x_0}(y) = \exp\left(-\frac{(I(x_0) - I(y))^2}{2\sigma_{x_0}^2}\right), \tag{1}$$

where $I$ is the image data, $y$ is a voxel in the volume and $\sigma$ is the mean square deviation of the pixels in the neighborhood of the start position from $x_0$. At any time of the random walks, the probability to move from a voxel $x$ to an adjacent voxel $y^{(j)}, j = 1, \dots, 6$, six-connected voxels, is given by

$$P_{x_0}\left(y^{(j)}, x\right) = \frac{w_{x_0}(y^{(j)})}{\sum_{i=1}^{6} w_{x_0}(y^{(i)})}. \tag{2}$$

For small $\sigma$, it is more likely that the random walks stay in an area similar to the neighborhood of the start position. With increasing $\sigma$, the weights and thus the probabilities for each direction approximate each other.

**Multi-GPU**. Biomedisa uses two approaches to perform random walks using Multi-GPU programming. In the first approach, the pre-segmented slices are distributed to as many GPUs as possible. In the second approach, the volume is decomposed into as many blocks as GPUs are available and the pre-segmented slices in the respective blocks are processed on the assigned GPU.

In the first approach, the hits of random walks must be stored for each label and for each GPU. They are stored in an array $n$ times the size of the image, where $n$ is the number of labels, including the background label. After all random walks have been calculated, the hits of a voxel computed by different GPUs are cumulated. For each voxel, the label with the highest number of hits is chosen as the final assignment. This is only possible for a small number of labels and is limited by the image size of the volume. The second approach is to divide the image data equally into blocks, with the number of blocks equaling the number of GPUs, so that each GPU calculates only the random walks that start in the pre-segmented slices of the assigned block. The random walks are not limited to the volume of the block, but can also exceed its limits. Therefore, so-called ghost blocks are attached that overlap with the neighboring blocks. After calculating the random walks for each label, the hits in the ghost blocks are sent to the GPUs of the neighboring blocks. Although Biomedisa's random walks can cover large distances, the risk of an incorrect segmentation at larger distances is high due to the decreasing density of random walks. We have found that a spacing of 100 slices is appropriate for the size of the ghost blocks.

Multiprocessing with several GPUs is realized with OpenMPI. Scalability tests were performed with 4 NVIDIA Tesla V100 on several datasets (Supplementary Fig. 14b). The method scales well with an increasing number of GPUs. For four GPUs, the speedup ranges from 46% (theropod claw) to 71% (head of the bull ant

queen) of the theoretical maximum. On average, the performance scales here by 60% of the theoretical maximum.

**Scalability of random walks**. Since it is difficult to deactivate individual compute units on a GPU, we use PyOpenCL[48] and 28 CPUs to demonstrate the scalability of the random walks on an image of the mouse molar teeth dataset (Supplementary Fig. 14c). We found a strong scalability of the random walks, where the speedup is 87% of the theoretical maximum for 7 CPUs, 77% for 14 CPUs and 68% for 28 CPUs.

**Smoothing**. Let $\Phi$ be the number of hits by random walks. The development of $\Phi$ by means of the partial differential equation

$$\frac{\partial \Phi}{\partial t} = \mu |\nabla \Phi| \nabla \cdot \left(\frac{\nabla \Phi}{|\nabla \Phi|}\right) \tag{3}$$

smoothes the topology of the hits and consequently the resulting segmentation, where the user-defined $\mu$ determines the magnitude of the change.

**Uncertainty**. The uncertainty of the segmentation at position $x$ is determined by

$$U(x) = 1 - \prod_{i \neq \max}\left(1 - \frac{\Phi_i(x)}{\Phi_{\max}(x)}\right), \tag{4}$$

where $\Phi_1, \dots, \Phi_n$ are the number of hits coming from $n$ different labels and $\Phi_{\max}$ is the maximum of all $\Phi_i$. The uncertainty takes values in the range of $0-1$, where 0 means no uncertainty and 1 high uncertainty, i.e. a voxel can be assigned to at least two labels with the same probability.

**Remove outliers and fill holes**. To remove possible outliers (unconnected voxels or islands), a technique has been integrated that first detects all distinct objects of the segmentation and then removes objects smaller than a predefined threshold (by default 90% of the size of the largest distinct object). Introducing a threshold for which objects are deleted prevents large objects which should be preserved from being deleted. The same technique is used to fill holes in objects. Here, holes are filled if they are smaller than a predefined threshold (by default 90% of the size of all objects belonging to the same label).

**Active contours**. The segmentation result is automatically post-processed with active contours[21] and the result is offered as an optional segmentation result. Active contours can also be used as a standalone segmentation method. In this case, the number of predefined iteration steps should be increased.

**Labeling in all axes**. The pre-segmentation is typically done using the standard orientation, with the $xy$-plane horizontal and the $z$-axis pointing up. If the standard orientation is not used exclusively, the function All axes must be activated in the settings.

**Two-network strategy and configuration**. Biomedisa uses Keras with TensorFlow backend to train a 3D U-Net[44]. A patch-based approach is used and can be combined with a two-network strategy in which a second network locally refines the result of the first network to compensate for errors caused by scaling large images. Similar to the training of the first network (see main text), the second network is trained by selecting a set of image files along with the corresponding fully segmented label files and here additionally with the first network. After training, the segmentation is performed by selecting an image together with both networks. For large image data, a rough approximation is made with the first network and refined with the second. The size of the patches is $64 \times 64 \times 64$ voxels for both networks. An overlapping of the patches is achieved by a stride size of e.g. 32 pixels that can be changed in Biomedisa. The patches serve as training data for a 3D U-Net. The network architecture of both networks follows the typical architecture of a 3D U-Net. It consists of a contracting and an expansive part with a repeated application of two $3 \times 3 \times 3$ convolutions, each followed by batch normalization and a rectified linear unit (ReLU) activation layer. Each contracting block is followed by a $2 \times 2 \times 2$ max pooling operation with stride 2 for downsampling. At each downsampling step, the number of feature channels is doubled, starting with 32 channels. Every step in the expansive part consists of an upsampling of the feature map and a concatenation with the corresponding cropped feature map from the contracting path, followed by two $3 \times 3 \times 3$ convolutions, with each followed by batch normalization and an ReLU activation layer. At the final layer, a $1 \times 1 \times 1$ convolution is used to map each feature vector to the desired number of classes. To train the network, stochastic gradient descent is used with a learning rate of 0.01, decay of $1 \times 10^{-6}$, momentum of 0.9, enabled Nesterov momentum, 200 training epochs, and a batch size of 24. All images are scaled to have the same mean and standard deviation.

**Evaluation of neural network**. To test the deep neural network integrated in Biomedisa, we used ten 3D cardiovascular magnetic resonance images from the HVSMR 2016 challenge as training data, which were manually segmented by

experts (see "Description of samples"). Using the trained network, the segmentation results from ten additional images (whose ground truth labels were not made public) were evaluated online on the challenge platform. On average, we achieved Dice scores of 0.762 ± 0.098 for the myocardium and 0.920 ± 0.016 for the blood pool. Here, refinement did not improve the accuracy so only the first network was trained. We used Biomedisa's feature to consider the voxel location, which takes into account the coordinates of the patches in the volume. To generate the patches, we used a stride size of 32 pixels. Before the training, the images were normalized and scaled to a size of 256 × 256 × 256 voxels. The training time was 7.32 h on 4 NVIDIA Tesla V100. The automatic segmentation of the test data took on average 1.37 min, using a stride size of 16 pixels and removing outliers with a threshold of 0.9 (see "Remove outliers and fill holes").

In addition, we used ten expert semi-automatically segmented μCT scans of mouse molar teeth (see "Description of samples") to test the model. The teeth were originally segmented using Biomedisa's semi-automatic diffusion method to study the effect of masticatory function on tooth enamel and dentine in adult mouse molars[41]. Here, three separate materials were segmented: enamel, dentine and alveolar bone (Fig. 5b, without surrounding alveolar bone tissue). The images have an average size of 438 × 543 × 418 voxels. We used a fivefold cross-validation where in each step eight images were used as training data and two images were retained as validation data for testing the model. On average, Dice scores of 0.949 ± 0.006 for the enamel, 0.982 ± 0.002 for dentine and 0.983 ± 0.002 for alveolar bone were achieved. By refining the result with a second network, results improved by 1.2% in enamel and 0.5% in both dentine and alveolar bone. Outliers were removed with a threshold of 0.05 (see "Remove outliers and fill holes"). The threshold was chosen to be very low to avoid deleting a dentine fragment consisting of several parts. Again, the images were normalized before the training and scaled to a size of 256 × 256 × 256 voxels. We used a stride size of 32 pixels for the first network and a stride size of 64 pixels for the second network. The average training time was 5.82 h on 4 NVIDIA Tesla V100 for the first network and 3.13 h for the second network. After training, the automatic segmentation of the test images took on average 34 s without refinement and 59 s with refinement. The fast evaluation enables the processing of a large number of samples.

**Evaluation metrics**. For two segmentations $X$ and $X'$ consisting of $n$ labels, the Dice similarity coefficient (Dice) is defined as

$$\text{Dice} = \frac{2\sum_{i=1}^{n}|X_i \cap X'_i|}{|X| + |X'|}, \quad (5)$$

where $|X|$ and $|X'|$ are the total number of voxels of each segmentation, respectively, and $X_i$ is the subset of voxels of $X$ with label $i$. For the boundaries $B$ of the pre-segmented slices and the surfaces $S'$ of the segmentation result, consisting of $n$ labels, the average surface distance (ASD) is defined as

$$\text{ASD} = \frac{1}{|B|}\sum_{i=1}^{n}\left(\sum_{p\in B_i} d(p, S'_i)\right), \quad (6)$$

where $|B|$ is the total number of points on the boundaries and

$$d(p, S'_i) = \min_{p'\in S'_i}\|p - p'\|_2 \quad (7)$$

is the Euclidean distance from a point $p$ on the boundary of label $i$ to the closest point on the corresponding surface of the segmentation result.

**Description of samples**

*Trigonopterus (Figs. 1–4, Supplementary Fig. 1).* Papuan weevil (*Trigonopterus* sp. (Coleoptera: Curculionidae) from the collection of the State Museum of Natural History Karlsruhe (volume size: 1497 × 734 × 1117 voxels). It was fixed in 100% ethanol and scanned in its defensive position at the UFO imaging station of the KIT light source (see Supplementary Table 2 for scan parameters). Sixty-four individual body parts were manually pre-segmented every fifth slice (=215 slices) using Amira 5.6. Different quantities of these pre-segmented slices were used as an input (Fig. 3) to evaluate the efficiency of the Biomedisa algorithm. For comparison, a conventional morphological interpolation was performed between the pre-segmented slices followed by manual correction (see main text). The computation for 37 selected pre-segmented slices took 19.72 min on 1 NVIDIA Tesla V100 and 10.37 min on 4 NVIDIA Tesla V100.

*Medaka (Fig. 5a, Supplementary Figs. 12, 13).* Japanese rice fish (*Oryzias latipes*) stained with 0.33% phosphotungstic acid (PTA) and 0.3% Lugol's iodine ($I_3K$) and embedded in 4% agarose (volume size: 900 × 1303 × 4327 voxels). Animal husbandry and experimental procedures were performed at the Institute of Biological and Chemical Systems (IBCS) of Karlsruhe Institute of Technology (KIT) in accordance with German animal protection regulations (Regierungspräsidium Karlsruhe, Germany; Tierschutzgesetz 111, Abs. 1, Nr. 1, AZ35-9185.64/BH). The IBCS is under the supervision of the Regierungspräsidium Karlsruhe, who approved the experimental procedures. The specimen was scanned with the laboratory X-ray setup of KIT's Institute for Photon Science and Synchrotron Radiation (see Supplementary Table 2 for scan parameters). Eleven individual body parts were pre-segmented in 231 slices. The different body parts were not always pre-segmented in the same slices. Therefore, each label was computed separately

resulting in computation times between 4.80 and 57.50 min on 1 NVIDIA Tesla V100 and between 4.53 and 24.23 min on 4 NVIDIA Tesla V100.

*Mouse molar teeth (Fig. 5b, Supplementary Figs. 4, 5).* Ten mouse mandibles were scanned ex-vivo on a diondo d3 μCT equipment (volume size on average: 438 × 543 × 418 voxels) and featured in a recent study[41]. On average 20 slices were pre-segmented in Avizo 9.2. A threshold-based selection with a manual correction was used to differentiate and pre-segment the enamel, dentine, and alveolar bone. Subsequently, 3D stacks were exported and processed semi-automatically with Biomedisa to segment the three materials. The computation took an average of 143.5 ± 11.32 s on 1 NVIDIA Tesla V100 and 53.6 ± 4.2 s on 4 NVIDIA Tesla V100. The data were also used to evaluate the deep neural network implemented in Biomedisa (see "Evaluation of neural network").

*Human hearts (Fig. 5c, Supplementary Figs. 6, 7).* Twenty 3D cardiovascular magnetic resonance (CMR) images were acquired during clinical practice at Boston Children's Hospital, Boston, MA, USA. Cases include a variety of congenital heart defects. Imaging was done in an axial view on a 1.5T Philips Achieva scanner (TR = 3.4 ms, TE = 1.7 ms, $\alpha$ = 60°) without contrast agent using a steady-state free precession pulse sequence. The data were provided by the organizers of the MICCAI Workshop on Whole-Heart and Great Vessel Segmentation from 3D Cardiovascular MRI in Congenital Heart Disease (HVSMR 2016, http://segchd.csail.mit.edu/) and featured by Pace et al.[46]. Ten training (including ground truth labels) and ten test CMR scans were provided either as a complete axial CMR image, as the same image cropped around the heart and thoracic aorta, or as a cropped short axis reconstruction. The ground truth labels were made by a trained observer and validated by two clinical experts. Image dimension and image spacing varied across subjects, and average 157 × 216 × 167 and 0.82 × 0.82 × 0.87 mm³, respectively. Two objects, the myocardium and the blood pool, were segmented. We used the ten cropped training images around the heart and thoracic aorta to evaluate Biomedisa. Using every 20th pre-segmented slice as initialization, the computation time averaged 3.4 ± 2.37 s on 1 NVIDIA Tesla V100 and 2.2 ± 1.08 s on 4 NVIDIA Tesla V100. In addition, the training and test images were used to evaluate the deep neural network (see "Evaluation of neural network").

*Wasp from amber (Fig. 5d, Supplementary Figs. 8, 9).* A ceraphronoid wasp (*Conostigmus talamasi* (Hymenoptera: Ceraphronidae)) trapped in an approx. 33 −55-million-year-old piece of Eocene Baltic amber (volume size: 1417 × 2063 × 2733 voxels)[39]. The specimen is stored at Senckenberg Deutsches Entomologisches Institut (Müncheberg, Germany) with collection number DEI-GISHym31819. Due to the size of the sample, it was scanned in three steps (upper, middle and lower part) at the UFO imaging station of the KIT light source (see Supplementary Table 2 for scan parameters). These scans were then combined into a single volumetric image. Fifteen individual body parts were pre-segmented every 20th slice resulting in 126 pre-segmented slices and a computation time of 79.28 min on 1 NVIDIA Tesla V100 and 31.80 min on 4 NVIDIA Tesla V100.

*Cockroach (Fig. 5e).* Dwarf hissing cockroach (*Elliptorhina chopardi* (Blattodea: Blaberidae)) fixed in 100% ethanol (volume size: 613 × 606 × 1927 voxels). The specimen was scanned with the laboratory X-ray setup of KIT's Institute for Photon Science and Synchrotron Radiation (see Supplementary Table 2 for scan parameters). A single label representing the tracheal network was pre-segmented every 25th slice resulting in 55 pre-segmented slices. The computation took 8.38 min on 1 NVIDIA Tesla V100 and 3.28 min on 4 NVIDIA Tesla V100.

*Theropod claw (Fig. 5f, Supplementary Figs. 10, 11).* Claw of an unknown juvenile theropod dinosaur (Theropoda: Coelurosauria) trapped in Burmese amber from the collection of Patrick Müller, Käshofen (volume size: 1986 × 1986 × 3602 voxels). It was scanned in two steps (upper and lower part) at the UFO imaging station of the KIT light source (see Supplementary Table 2 for scan parameters). A single label representing the claw was pre-segmented every 40th slice resulting in 88 pre-segmented slices. The computation took 29.50 min on 1 NVIDIA Tesla V100 and 16.03 min on 4 NVIDIA Tesla V100.

*Mineralized wasp (Fig. 5g, Supplementary Figs. 2, 3).* An approx. 34−40-million-year-old parasitoid wasp (*Xenomorphia resurrecta* (Hymenoptera: Diapriidae)) preserved inside a mineralized fly pupa from the Paleogene of France (volume size: 1077 × 992 × 2553 voxels). The specimen is stored at the Natural History Museum of Basel with collection number NMB F2875. It was scanned in two steps (upper and lower part) at the UFO imaging station of the KIT light source (see Supplementary Table 2 for scan parameters). Fifty-six individual labels were pre-segmented every tenth slice resulting in 223 pre-segmented slices and a computation time of 55.95 min on 1 NVIDIA Tesla V100 and 21.77 min on 4 NVIDIA Tesla V100. It was featured in a recent study along similar specimens that were processed in the same way[42].

*Bull ant queen (Fig. 5h).* The head of an Australian bull ant queen (*Myrmecia pyriformis* (Hymenoptera: Formicidae)) was scanned at the UFO imaging station of the KIT light source (see Supplementary Table 2 for scan parameters, volume size: 1957 × 1165 × 2321 voxels). It should be noted that the long exposure time was not

optimal for this kind of specimen and resulted in bubble formation in the soft tissue, subsequent artifacts in the tomogram and generally noisy data. Artifacts caused by diffuse edges in the tomogram required slight manual corrections. Fifty-two individual body parts were pre-segmented. The head, left antenna, and right antenna were segmented separately. The distance between the pre-segmented slices was chosen between 5 and 50, depending on the morphology. For the head, 81 slices were pre-segmented, 109 for the left antenna and 114 for the right antenna. The computation took 42.25, 11.17 and 16.87 min on 1 NVIDIA Tesla V100 and 14.90, 5.40 and 6.68 min on 4 NVIDIA Tesla V100, respectively. The final surface mesh was artificially colored and animated using CINEMA 4D R20.

*Human mandibles (Fig. 7).* Computed tomography datasets of the craniomaxillofacial complex were collected during routine clinical practice in the Department of Oral and Maxillofacial Surgery at the Medical University of Graz in Austria[47]. Ten CT scans were selected and complete human mandibles were segmented by two clinical experts in MeVisLab[52]. The segmentations were saved as contour segmentation objects (CSO) files. In order to process the data with Biomedisa, CSO files were converted to the NRRD file format with MeVisLab. CSO were loaded with CSOLoad and corresponding CT data with itkImageFileReader. Both served as input to CSOConvertToImage. Finally, the converted image was saved with itkImageFileWriter as NRRD. The CT scans were performed with a Siemens Somatom Sensation 64 medical scanner (Dose of Scan = 120 kV, Scan Exposure = 285.5 mAs (on average). Image dimension and image spacing are on average $512 \times 512 \times 195$ and $0.453 \times 0.453 \times 1.3$ mm$^3$, respectively. Using approximately every 20th slice of the manually labeled data as initialization (which corresponds to 5−8 pre-segmented slices), the average computation time was $79.0 \pm 15.52$ s on 1 NVIDIA Tesla V100 and $31.6 \pm 3.62$ s on 4 NVIDIA Tesla V100.

### Post-processing

*Tomographic data.* Tomographic reconstructions were performed with a GPU-accelerated filtered back projection algorithm implemented in the UFO software framework[53]. µCT reconstructions were converted from 32-bit to 8-bit in order to reduce data size and improve handling. If the sample was scanned in several parts due to its size, the resulting tomographic volumes were registered and merged with Amira (version 5.6, FEI). All datasets were realigned and cropped to optimize sample orientation and position inside the volume. After processing with Biomedisa, the result of medaka was downscaled from $900 \times 1303 \times 4327$ voxels to $450 \times 651 \times 2163$ voxels to facilitate further processing.

*Segmentation results.* Outliers were removed, all labels converted into polygon meshes with Amira 5.6, exported as OBJ files and reassembled in CINEMA 4D R20. CINEMA 4D R20 was used for smoothing, polygon reduction and for rendering of the figures and animations.

*Uncertainty quantification.* The volume renderings shown in Fig. 4 were done in Amira 2019.2.

**Reporting summary**. Further information on research design is available in the Nature Research Reporting Summary linked to this article.

### Data availability

Datasets *Trigonopterus* (Figs. 1–4), mouse molar teeth (Fig. 5b), wasp from amber (Fig. 5d), cockroach (Fig. 5e), theropod claw (Fig. 5f), mineralized wasp (Fig. 5g) and bull ant queen (Fig. 5h) are available at https://biomedisa.info/gallery. Dataset human hearts (Fig. 5c) is from the MICCAI Workshop on Whole-Heart and Great Vessel Segmentation from 3D Cardiovascular MRI in Congenital Heart Disease (HVSMR 2016). Information on how to obtain the data can be found at http://segchd.csail.mit.edu. Dataset human mandibles (Fig. 7) can be downloaded at https://doi.org/10.6084/m9.figshare.6167726.v5. Further data will be made available from the corresponding author upon reasonable request.

### Code availability

The source code is freely available as part of the open-source software Biomedisa. It was developed and tested for Ubuntu 18.04 LTS and Windows 10. Any common browser can be used as an interface. Biomedisa can be downloaded at https://github.com/biomedisa/biomedisa and installed according to the installation instructions.

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

## Acknowledgements

Biomedisa was developed in the scope of the projects ASTOR and NOVA, funded by the German Federal Ministry of Education and Research (BMBF; 05K2013 and 05K2016). We acknowledge the support by the projects UFO 2 (BMBF; 05K2012), CODE-VITA (BMBF; 05K2016) and HIGH-LIFE (BMBF; 05K2019), the state of Baden-Württemberg through bwHPC, the Ministry of Science, Research and the Arts Baden-Württemberg (MWK) through the data storage service SDS@hd, and the German Research Foundation (DFG; INST 35/1314-1 FUGG and INST 35/1134-1 FUGG). We especially thank Alexander Riedel for providing the *Trigonopterus* weevil and taking the photograph of the original specimen. We are grateful to Julián Balanta-Melo, Hartmut Greven, Felix Loosli, István Mikó, Patrik Müller and Arnold Staniczek for providing sample specimens, Felix Beckmann, Philipp Gerstner, Jörg Hammel, Wolfgang Mexner, Venera Weinhardt, Martin Wlotzka, Peter Zaspel and Yaroslav Zharov for helpful comments and fruitful discussions, and Stephen Doyle for improving the language of the manuscript. We acknowledge the KIT light source for provision of instruments at their beamlines and we would like to thank the Institute for Beam Physics and Technology (IBPT) for the operation of the storage ring, the Karlsruhe Research Accelerator (KARA).

## Author contributions

P.D.L., T.v.d.K., M.H., A.K., J.W., T.B. and V.H. conceived and designed the study. P.D.L. developed Biomedisa and devised algorithms. T.v.d.K., N.A., S.B., J.O. and M.Z. performed original μCT scans. P.D.L., T.v.d.K., N.A., J.O. and S.S. analyzed and pre-segmented data. A.J. contributed to the development of Biomedisa. P.D.L., T.F. and A.E. tested the platform and carried out the evaluation. O.P. performed security tests and developed the automatic installation. N.T.J. developed the web-based visualization tool. S.A.C. tested virtualization techniques. T.v.d.K. created renderings and animations. V.H. supervised the project. P.D.L. and T.v.d.K. wrote the manuscript. All authors contributed to the writing and discussion.

## Funding

## Competing interests

The authors declare no competing interests.
