## [Peer Review File · Nature Communications]

Reviewers' comments:

Reviewer #1 (Remarks to the Author):

The present work presents a software package of innovative and urgently needed segmentation tools for 3d data annotation and segmentation, with a comprehensive description and convincing example. It is likely to meet the needs of the field (biomedical 3d image analysis in particular CT and MRI), and I would like to congratulate the authors to their work.

Already by brief inspection of the software and metadata, I am convinced of its usefulness and soundness. I would also be interested in further testing on different problems, but this will take more time than available for the review.

Independently of this, I am unsure whether the work is suitable for publication in Nature Communications or Nature Methods (which in my view would fit better in scope). Since the underlying algorithm seems to be identical or at least similar to the one of Ref.18., the novelty of the present work as a piece of research has to be better explained in order to meet the level which I would expect for these particular journals. There are probably also a number of extensions and algorithmic improvements in the random walk approach, but the emphasis seems to be on the implementation, integration and testing of the software tools. In the case that the algorithm is essentially the same as in Ref.18, the suitability of the manuscript has to be judged as a scientific software contribution more than a particular research project in image analysis. Whether this can still be suitable for Nat.Comm. must then be an editorial decision. All I can say, is that I find the work relevant, sound and well described. It certainly merits a good publication to warrant citability and scientific accountability. Note that software-associated publications can reach an outstanding number of citations, even if not published in a journal of high impact factor.

Reviewer #2 (Remarks to the Author):

Key results: This manuscript details an open source web application for segmentation of 3D image datasets. The method presented significantly reduces the segmentation time particularly from the perspective of user input. The method presented is applicable to a wide range of 3D imaging datasets and the open source platform is accessible. Further the description of the segmentation protocol is clear and a variety of examples are presented demonstrating the utility of the methodology.

Validity: The manuscript does not appear to be flawed.

Originality and significance: The manuscript would be strengthened through inclusion of a critical review of current techniques in comparison to the proposed method. Although the manuscript does contain some comparisons additional detail and critic would be welcome. A more detailed review would also assist in demonstrating the originality of the work. The proposed methodology appears to be a robust combination of existing methodologies and a description of how this is significantly original would be helpful. In the main text reference to deep neural networks is made and the significant requirement for large amounts of training data is noted as a weakness. Have the authors considered transfer learning approaches which are less demanding in terms of training and user input? Overall this is a sound piece of work however the manuscript as it stands does not draw strong conclusions nor does it provide a critical analysis of the proposed method in the context of competing techniques.

Data & methodology: The approach presented appears valid and sufficient detail is provided to enable reproducibility of results.

Appropriate use of statistics and treatment of uncertainties: Error bars are perhaps not relevant given the nature of the 3D data reconstructions. The authors do specify errors associated with segmentation and specify accuracy metrics based on the Dice similarity coefficient.

Conclusions: The data interpretation is clear.

Suggested improvements: I recommend the authors revise the manuscript to clearly articulate the novelty of the presented method. At present it appears that existing methods are combined to reduce segmentation time. Whilst this is a solid piece of work it at presented does not seem to be a significantly novel approach. Thus, the authors are encouraged to address this point.

References: The manuscript reference previous literature appropriately.

Clarity and context: The abstract is clear and accessible. The introduction and conclusions are lacking, particularly the conclusion. The conclusion should be compelling and critical.

Reviewer #3 (Remarks to the Author):

In this paper, an image segmentation framework called Biomedisa is presented. The approach uses random walk algorithm to connect pre-segmented 2D slices to achieve 3D volume data segmentation. Although the presented example looks impressive, the paper has a couple of major drawbacks.

1. No review of related works

Given the large body of literatures on biomedical image segmentation, the authors should provide a brief review to state clearly the research gap. What are the advantages and disadvantages of the proposed method? There have been thousands of image segmentation methods, including many excellent works for biological image segmentation. What is special about this one?

2. No quantitative evaluation of the method

Only one example is given to demonstrate the effect of the proposed method. There is no quantitative evaluation against other methods. A very brief evaluation regarding the influence of the number of pre-segmented slices are given. It is not clear how the quality of such pre-segmentation will affect the overall segmentation performance.

3. No evaluation against other state-of-the-art methods

Again, only a single case is presented, without comparing to any other methods. Given the advances of the image segmentation field, this is unacceptable. The authors should compare their method to a number of state-of-the-art methods over a number of large datasets to support their strong claim about the applications of the proposed method.

We thank all reviewers for their constructive feedback. The main concerns referred to 1) novelty of the presented research, 2) short or lacking introduction and conclusion and 3) an insufficient review of and lacking comparison with current state-of-the-art methods.

The reviewers' remarks encouraged us to revise the paper substantially. The new manuscript emphasizes Biomedisa's unique capabilities as an easy-to-use online platform that addresses the needs of scientists without substantial computational expertise more clearly and includes comprehensive introduction and discussion sections. Further, we evaluated Biomedisa's performance against other methods and provide corresponding figures, tables and supplementary material. We demonstrate that Biomedisa's segmentation performance is superior to the most common approach of morphological interpolation and in almost all scenarios to all other evaluated methods with respect to both speed and accuracy.

List of most important changes:

- The paper was fundamentally extended, now including more comprehensive introduction and discussion sections and a detailed evaluation against other methods.
- Fig. 1 was updated to improve comprehensibility
- Table 1, Figs 6 & 7 and Suppl. Table 1 and Suppl. Figs 1-14 (containing cross-sections of the results in combination with the segmentation errors) were added to support the evaluation.
- Sample datasets "human mandibles" were added to evaluate the robustness of Biomedisa against input errors.
- Title was slightly changed to "Biomedisa: an open-source online **platform** for biomedical image segmentation" in order to emphasize the uniqueness as a platform solution
- A. Ershov and T. Faragó were added to the list of authors
- Reference list was extended
- The paper was formatted according to the style of Nature Communications

Please find below our answers to the specific remarks raised by the three reviewers.

Reviewers' comments:

Reviewer #1 (Remarks to the Author):

The present work presents a software package of innovative and urgently needed segmentation tools for 3d data annotation and segmentation, with a comprehensive description and convincing example. It is likely to meet the needs of the field (biomedical 3d image analysis in particular CT and MRI), and I would like to congratulate the authors to their work.

Already by brief inspection of the software and metadata, I am convinced of its usefulness and soundness. I would also be interested in further testing on different problems, but this will take more time than available for the review.

Independently of this, I am unsure whether the work is suitable for publication in Nature Communications or Nature Methods (which in my view would fit better in scope). Since the underlying algorithm seems to be identical or at least similar to the one of Ref.18., the

novelty of the present work as a piece of research has to be better explained in order to meet the level which I would expect for these particular journals. There are probably also a number of extensions and algorithmic improvements in the random walk approach, but the emphasis seems to be on the implementation, integration and testing of the software tools. In the case that the algorithm is essentially the same as in Ref.18, the suitability of the manuscript has been judged as a scientific software contribution more than a particular research project in image analysis. Whether this can still be suitable for Nat.Comm. must then be an editorial decision. All I can say, is that I find the work relevant, sound and well described. It certainly merits a good publication to warrant citability and scientific accountability. Note that software-associated publications can reach an outstanding number of citations, even if not published in a journal of high impact factor.

The references 16 & 18 (36 & 40 in the revised version) feature earlier developmental versions of the algorithm published in conference proceedings, which were not as advanced as the Biomedisa online platform presented here. Since then, the algorithm was significantly improved, including automatically adapting of the number of random walks and embedding the calculation of the random variables, which has dramatically increased the speed of the segmentation process.

Moreover, many essential features were added, including a specifically developed smoothing technique and an uncertainty quantification of the segmentation result, which were both embedded in the computation of the random walks, and the training of neural networks.

The developmental versions of the code were not published before. This paper makes both the platform and the underlying source code accessible to the scientific community for the first time.

Our work was motivated to a large extent by the goal to provide an unique easy-to-use software tool to a broad user community. However, in the revised version we strengthened the research aspect of the article by providing a critical review of current techniques and an evaluation of Biomedisa against other methods, thus highlighting its advantages and scientific relevance.

Reviewer #2 (Remarks to the Author):

Key results: This manuscript details an open source web application for segmentation of 3D image datasets. The method presented significantly reduces the segmentation time particularly from the perspective of user input. The method presented is applicable to a wide range of 3D imaging datasets and the open source platform is accessible. Further the description of the segmentation protocol is clear and a variety of examples are presented demonstrating the utility of the methodology.

Validity: The manuscript does not appear to be flawed.

Originality and significance: The manuscript would be strengthened through inclusion of a critical review of current techniques in comparison to the proposed method. Although the manuscript does contain some comparisons additional detail and critic would be welcome. A more detailed review would also assist in demonstrating the originality of the work.

The proposed methodology appears to be a robust combination of existing methodologies and a description of how this is significantly original would be helpful.

In the main text reference to deep neural networks is made and the significant requirement for large amounts of training data is noted as a weakness. Have the authors considered transfer learning approaches which are less demanding in terms of training and user input? Overall this is a sound piece of work however the manuscript as it stands does not draw strong conclusions nor does it provide a critical analysis of the proposed method in the context of competing techniques.

Data & methodology: The approach presented appears valid and sufficient detail is provided to enable reproducibility of results.

Appropriate use of statistics and treatment of uncertainties: Error bars are perhaps not relevant given the nature of the 3D data reconstructions. The authors do specify errors associated with segmentation and specify accuracy metrics based on the Dice, additional features, similarity coefficient.

Conclusions: The data interpretation is clear.

Suggested improvements: I recommend the authors revise the manuscript to clearly articulate the novelty of the presented method. At present it appears that existing methods are combined to reduce segmentation time. Whilst this is a solid piece of work it as presented does not seem to be a significantly novel approach. Thus, the authors are encouraged to address this point.

References: The manuscript reference previous literature appropriately.

Clarity and context: The abstract is clear and accessible. The introduction and conclusions are lacking, particularly the conclusion. The conclusion should be compelling and critical.

The manuscript was substantially extended and now includes a paragraph dedicated to current segmentation techniques, their strengths and weaknesses, and highlights the originality of Biomedisa as an easy-to-use one-click online solution that addresses the needs of scientists even without computational expertise.

Additionally, we emphasize the improvements of the algorithm and extensions of the Biomedisa online platform (see also our answers to reviewer 1).

Transfer learning, like refining a pre-trained neural network, can reduce both training time and training data. However, the focus of our study is on supporting the manual segmentation when there is little a priori information available, e.g. for the dense annotation of the training data for a deep neural network. Therefore, the emphasis of our paper is the semi-automated segmentation by Biomedisa's random walk algorithm. But we agree that transfer learning would be a very attractive additional feature for Biomedisa and we plan to add this as a module in the future.

We added a comprehensive evaluation of Biomedisa, where we compared Biomedisa with existing established methods (Graph Cut, Random Walker, GoeS, morphological interpolation of ITK and Amira).

Further, we have significantly expanded the introduction and critical discussion (there is no dedicated conclusion paragraph in Nature Communications).

Reviewer #3 (Remarks to the Author):

In this paper, an image segmentation framework called Biomedisa is presented. The approach uses random walk algorithm to connect pre-segmented 2D slices to achieve 3D volume data segmentation. Although the presented example looks impressive, the paper has a couple of major drawbacks.

1. No review of related works

Given the large body of literatures on biomedical image segmentation, the authors should provide a brief review to state clearly the research gap. What are the advantages and disadvantages of the proposed method? There have been thousands of image segmentation methods, including many excellent works for biological image segmentation. What is special about this one?

2. No quantitative evaluation of the method

Only one example is given to demonstrate the effect of the proposed method. There is no quantitative evaluation against other methods. A very brief evaluation regarding the influence of the number of pre-segmented slices are given. It is not clear how the quality of such pre-segmentation will affect the overall segmentation performance.

3. No evaluation against other state-of-the-art methods

Again, only a single case is presented, without comparing to any other methods. Given the advances of the image segmentation field, this is unacceptable. The authors should compare their method to a number of state-of-the-art methods over a number of large datasets to support their strong claim about the applications of the proposed method.

We significantly expanded the introduction and discussion sections of the paper, now providing a comprehensive overview regarding state-of-the-art methods and their suitability for different scenarios. We now clearly state the advantages of Biomedisa for segmenting large image data with little a priori knowledge and emphasize its accessibility as an easy-to-use one click solution for users without in-depth IT knowledge.

The revised manuscript now contains a quantitative evaluation of Biomedisa's primary segmentation algorithm compared to different popular semi-automatic segmentation methods on a variety of large datasets as well as an evaluation of its robustness against input errors (Fig. 7).

The implementation of the compared tools for large images evaluated proved to be difficult and time-consuming even for an expert. Some tools (scikit-image (Random Walker) and medpy (Graph Cut)) are limited by the image size and do not work on large data. A specially developed block-by-block strategy was required to allow comparison with these tools. This experience further illustrated the need of an easy-to-use application of semi-automatic segmentation algorithms for scientists without computational background.

Reviewer #3 (Remarks to the Author):

The reviewer appreciates the efforts of the authors in addressing the previous raised concerns to the original submission. The paper has been significantly improved by having more results and quantitative comparison. However, the reviewer thinks that the core concern remains, which is regarding to the lack of novelty in methodology. This paper currently describes a software platform, which is important and interesting, but the contribution is more on implementation and development. The same concern is shared among the reviewers.

The mentioned technical contributions are more about software development. The newly added quantitative comparison is not completely fair. The developed software was compared to other development libraries, like Scikit-Image, which aims to provide researchers a tool as building blocks to develop their own applications.

Overall, the paper has been improved but not all the concerns have been addressed. Major concerns remain unsolved.

We thank reviewer #3 for the feedback. We appreciate the acknowledgement of the improvement of our manuscript. However, we do not agree with the assessment that "the contribution is more on implementation and development", since the contribution is primarily on presenting a new and unique online platform, its use and evaluation.

In addition, we disagree that "the quantitative comparison is not completely fair" as we compared Biomedisa with the most common method of manually segmenting large images (i.e., morphological interpolation, such as with Amira) and some of the most widely used techniques and software libraries.